# Evaluation of Performance of C-Reactive Protein (CRP) and Interferon-Gamma-Inducible Protein 10 (IP-10) as Screening for Active Tuberculosis

**DOI:** 10.3390/tropicalmed10110306

**Published:** 2025-10-27

**Authors:** Rotimi Samuel Owolabi, Russel Dacombe, Konstantina Kontogianni, Olusegun M. Akinwande, Lovett Lawson, Luis E. Cuevas

**Affiliations:** 1Liverpool School of Tropical Medicine, Pembroke Place, Liverpool L3 5QA, UK; russell.dacombe@lstmed.ac.uk (R.D.); nadia.kontogianni@lstmed.ac.uk (K.K.); olusegun.akinwande@lstmed.ac.uk (O.M.A.); luis.cuevas@lstmed.ac.uk (L.E.C.); 2University of Abuja Teaching Hospital (UATH), Gwagwalada, Abuja P.M.B. 228, Nigeria; 3Nigerian Institute of Medical Research (NIMR), Yaba, Lagos P.M.B. 2013, Nigeria; 4Zankli Research Centre, Bingham University, KM 26 Abuja-Keffi Expressway, Kodape, Karu P.M.B. 005, Nasarawa State, Nigeria; lovettlawson@hotmail.com

**Keywords:** active tuberculosis, screening, C-reactive protein, interferon-gamma-inducible protein 10, IP-10, Nigeria

## Abstract

Background: Most of the currently approved TB diagnostics are sputum-based. However, due to unusual clinical presentations of TB among HIV patients, they may not have TB symptoms and be able to produce sputum. Hence, these diagnostics may not be able to detect as many TB cases as possible among these patients. Therefore, this study assessed the performance of C-reactive protein (CRP) and interferon-gamma-inducible protein 10 (IP-10) as a screening tool for TB. Methods: This prospective study was conducted by consecutively recruiting patients with TB symptoms, collecting their sputum and blood samples, using sputum culture as the reference standard, and determining the best cut-off point of serum levels of CRP and IP-10 (separately and in combination) for TB diagnosis. Findings: CRP and IP-10 were measured in 408 patients with TB symptoms, of which 21% had culture-confirmed TB. CRP’s sensitivity and specificity were (91.4% and 33.2%), (95.3% and 42.6%) and (84.8% and 22.1%) for the whole study population, HIV-negative and HIV-positive patients, respectively. The sensitivity and specificity of IP-10 were (87.3% and 40.9%), (87.5% and 50.3%) and (79.4% and 47.2%) for the patients’ categories, respectively. Combination of CRP and IP-10 slightly improved the performance of the biomarkers among HIV-negative patients, with sensitivity of 97.5% and specificity of 43.3%. Interpretation: Though CRP and IP-10 performed better in HIV-negative patients than among people living with HIV (PLHIV), the performance of the biomarkers is lower than what is recommended by the WHO (sensitivity ≥ 90% and specificity ≥ 70%) for a TB screening tool. Hence, there is a need for better non-sputum-based TB diagnostics.

## 1. Introduction

Every year, millions of cases of tuberculosis (TB) are undetected globally due to the poor performance of the most commonly available TB diagnostics, particularly in high-burden settings. Of the 10.8 million estimated cases of TB in 2023, more than one third (2.7 million TB cases) were not diagnosed or reported to the World Health Organization (WHO) [1]. Although the sensitivity of smear microscopy is generally poor and worse (14–65%) among HIV patients [2], it still the most widely used diagnostic test in most high-burden settings because of its low cost and availability in local laboratories. New molecular automated diagnostic platforms with high sensitivity and specificity, which can detect Mycobacterium tuberculosis (MTB) and resistance to first line anti-TB drugs, are still costly, not available, or too complex to use at the lowest levels of healthcare delivery in most high-burden countries [3].

In addition, studies that assessed the performance of the WHO symptom screening algorithms to screen patients for TB and to rule out TB among HIV-infected individuals have shown that symptoms alone are unreliable, resulting in high false positive rates among PLHIV and missing many cases with TB. Undergoing further diagnostic workup by all patients with positive TB symptom screening will result in overloading the already weak laboratory capacity in most high-burden settings and delay commencement of preventive therapy for those without active TB [4,5].

In response to the challenge in TB diagnosis, the WHO convened a meeting in April 2014 to develop a consensus of four high-priority target product profiles (TPPs) for new TB diagnostics. These include: a non-sputum-based point-of-care (POC) test that can detect all forms of TB through identification of biomarkers or biosignatures; a POC triage test that is simple, not costly, and can be used to rule out TB at the lowest level of healthcare; a sputum-based POC test that could replace smear microscopy; and a rapid drug susceptibility test (DST) at the microscopy level [6]. Of these, the most urgently needed diagnostic is the non-sputum-based POC test that can accurately detect all forms of TB with a sensitivity higher than smear microscopy.

The need for better (simple and inexpensive) TB diagnostics or screening tools (in high-burden settings) has stimulated research efforts towards immunodiagnosis by looking for biomarkers that may be potential candidates for diagnosing or ruling out TB. Identifying biological markers which can be easily measured in the blood or other body fluid and accurately detect TB will greatly overcome the difficulty associated with TB diagnosis among HIV patients and be good for diagnosing extrapulmonary or culture-negative TB. Consequently, several biomarkers have been assessed as possible TB diagnostics or screening tools [7,8]. Two of them, C-reactive protein (CRP) and interferon-gamma-inducible protein 10 (IP-10), have potential as screening tests, although most studies to date have focused on their potential for TB diagnosis and treatment monitoring [8,9] and none of the studies that evaluated these biomarkers as screening tools for TB was conducted in Nigeria.

Furthermore, most of the current TB diagnostics are sputum-based. However, due to the unusual clinical presentation of TB among HIV patients, they may not have any TB symptoms and not be able to produce sputum [4]. Hence, these diagnostics may miss many TB cases and new approaches are needed to improve TB case detection among HIV-infected and uninfected patients. As there are no products that are close to market, this study evaluated the performance of C-reactive protein (CRP) and interferon-gamma-inducible protein 10 (IP-10) as biomarkers to rule out active TB.

## 2. Methodology

### 2.1. Study Design

This prospective study was conducted on patients with TB symptoms presenting at Asokoro District Hospital (ADH) and Nyanya General Hospital (NGH), both in Abuja, Nigeria, between 17 August 2017 and 8 January 2018.

### 2.2. Description of Study Area/Site

Abuja, in the Federal Capital Territory (FCT) of Nigeria, is located in the north-central geopolitical zone of the country and is a rapidly growing city, with a population of about 3.6 million (3,564,100). The FCT has 3 tertiary, 14 secondary and 179 public health facilities and 673 registered private health facilities across six area councils. HIV prevalence and TB/HIV coinfection rates in the FCT are 7.5% and 32.3%, respectively.

This study was conducted at the Directly Observed Treatment Short-course (DOTs) clinics of the hospitals. ADH is a 160-bedded hospital and serves about 250,000–500,000 residents of Abuja. The hospital provides services in all major specialties and comprehensive TB and HIV care. NGH is a 68-bedded hospital and serves people of lower socioeconomic status than ADH.

All patients attending the hospitals report to the General Outpatients Department (GOP), where each of them will be seen by a Medical Officer, who will take their clinical history. Patients with symptom(s) suggestive of TB are then referred to the DOTs clinic for TB screening. At the DOTs clinic, patients receive investigation forms for smear microscopy or Xpert MTB/RIF and chest X-rays. Thereafter, patients go to the laboratory for investigations and collect the investigations’ results at the DOTs clinic. Patients with a positive TB test are registered for treatment and counselled. The staff of the DOTs clinic ask about contacts and arrange for contact tracing.

Patients with drug-susceptible TB receive a six-month regimen, comprising a two-month intensive phase and a continuation phase of four months. The intensive phase consists of a fixed dose combination (FDC) of rifampicin, isoniazid, pyrazinamide and ethambutol. The continuation phase includes FDC of rifampicin and isoniazid. Patients with drug-resistant TB are referred for DST and to the drug-resistant TB treatment centres.

### 2.3. Patients’ Recruitment and Data Collection

Every consecutive patient with TB symptoms referred to the DOTs clinics was approached to participate in the study. After full informed consent, patients that agreed to participate were recruited by a trained community extension worker (CHEW), using a structured questionnaire. Two sputum samples were collected from each participant to conduct culture and Xpert MTB/RIF. A rapid CRP test was conducted at the DOTs clinic, using the Actim CRP (Medix Biomedica). This is a semiquantitative immunochromatographic dipstick test that is visually interpreted and its result can be read within five minutes. The test was carried out by trained CHEWs on whole blood collected from fingertips and interpreted according to its manufacturer’s instructions.

A 5 mL blood sample was collected for serum IP-10 and CRP immunoassays. Specimens were transported in a cold chain on a daily basis to Zankli Research Centre by a dedicated driver.

### 2.4. Inclusion Criteria

Age ≥ 18 years.

Cough for ≥2 weeks.

Consented to participate.

Agreed to provide the needed specimens’ samples (blood and sputum).

### 2.5. Exclusion Criteria

Currently on TB treatment.

Already diagnosed with TB but has not commenced TB treatment.

Patients with language barriers and no appropriate interpreter.

Patients willing to participate but who could not read/write and had no legally assigned representative to sign the consent form and act as a witness.

### 2.6. Laboratory Analysis of Samples

The first sputum sample was tested with Xpert MTB/RIF. The second sputum sample was decontaminated and digested, using sodium hydroxide- N-acetyl-L-cysteine-Citrate solution (Mycoprep reagent), for culture using solid culture in LJ medium. Culture- or Xpert-positive samples were classified as TB-positive, while those with both negative culture and Xpert results were classified as TB-negative.

The blood sample collected was analysed for serum IP-10 using an enzyme-linked immunosorbent assay (ELISA) (R&D systems, Inc., Minneapolis, MN, USA) that uses a quantitative sandwich enzyme immunoassay. The study flow chart is shown in Figure 1.

### 2.7. Ethical Considerations

Ethical approvals were obtained from the institutional Review Boards of the Liverpool School of Tropical Medicine (LSTM) (Reference number: 15-045, Date of approval: 15 February 2016), University of Abuja Teaching Hospital (UATH) (Reference Number: FCT/UATH/HREC/PR/491, Date of approval: 10 February 2016) and the Nigeria’s Federal Capital Territory (FCT) (Reference Number: FHREC/2016/01/76/17-10-16, Date of approval: 17 October 2016). Written informed consent was obtained from all participants.

### 2.8. Data Analysis

Data entry was conducted using Epi-Info 7 (Center for Disease Control and Prevention (CDC), Atlanta, GA, USA) and analysis of the data was performed in IBM SPSS (Version 25) Statistics (International Business Machine Corporation (IBM)). Descriptive statistics were run to obtain counts, percentages, medians and interquartile ranges, as appropriate. Thereafter, crosstabulations and receiver operating characteristic (ROC) curves were prepared to determine sensitivity and specificity of the CRP and IP-10 at various cut-off points and to ascertain the best cut-off point for the biomarkers.

## 3. Results

### 3.1. Participants’ Characteristics and Laboratory Investigations

A total of 408 patients were recruited and had blood samples collected for CRP and IP-10. One hundred and ninety-four (47.5%) were male and two hundred and fourteen (52.5%) female. Their median (interquartile range (IQR)) age was 36 (28.5–43.0) years.

All the patients had cough for >2 weeks, with a median (IQR) cough duration of 4 (2–6.5) weeks. Other frequent clinical symptoms were fever (271, 66.4%); weight loss (280, 68.6%); night sweats (210, 51.5%); chest pain (256, 62.7%); body weakness (233, 57.1%); loss of appetite (140, 34.3%) and haemoptysis (56,13.7%). Almost all the patients (388, 95.1%) had HIV testing results and 182 (46.9%) were positive. Sixty (14.7%) participants had been previously treated for TB and were being re-investigated (Table 1).

Nearly all (396, 97%) participants had sputum culture and 326 (82.3%) were culture-negative, with 62 (15.7%) being culture-positive and 8 (2%) contaminated. In addition, 407 (99.8%) patients had valid GeneXpert results. Rifampicin resistance was detected in seven (9.1%) patients. TB was confirmed by either culture or Xpert in 86 (21.1%) of the participants (Table 1).

Rapid CRP testing was available for 382 (93.6%) participants (Table 1).

### 3.2. Performance of CRP and IP-10 as a Screening Marker for TB

Three cut-off points of CRP (≥10 mg/L, ≥40 mg/L and >80 mg/L) were assessed using sputum culture as the reference standard. CRP ≥ 10 mg/L performed fairly well among the study population but had a poorer performance among HIV-positive patients (Table 2).

A CRP cut-off ≥ 40 mg/L had the highest performance among all patients, with the sensitivity, specificity, PPV and NPV being 72.8% (95% CI = 61.8–82.1%), 70.1% (95% CI = 64.6–75.2%), 39.6% (95% CI = 31.7–47.9%) and 90.6% (95% CI = 86.1–94.0%), respectively

The same cut-off point had the highest performance among HIV-positive patients, with sensitivity, specificity, PPV and NPV of 84.8% (95% CI = 68.1–94.9%), 22.1% (95% CI = 15.4–30.0%), 20.9% (95% CI = 14.4–28.8%) and 85.7% (95% CI = 69.7–95.2%), respectively.

However, a lower cut-off point of CRP ≥ 10 mg/L had the highest performance among HIV-negative patients, with sensitivity, specificity, PPV and NPV of 95.3% (95% CI = 84.2–99.4%), 42.6% (95% CI = 34.7–50.8%), 31.5% (95% CI = 23.7–40.3%) and 97.1% (95% CI = 89.8–99.6%), respectively.

The performance of IP-10 at six different cut-off points (>50 pg/mL, >100 pg/mL, >150 pg/mL, >200 pg/mL, >250 pg/mL and >300 pg/mL) is shown in Table 2.

For the whole study population, an IP-10 cut-off > 100 pg/mL had the highest performance with sensitivity of 87.3% (95% CI = 78.0–93.8%), specificity 40.9% (95% CI = 35.4–46.6%), PPV 27.2% (95% CI = 21.8–33.1%) and NPV 92.8% (95% CI = 87.1–96.5%).

The same cut-off point had the highest performance among HIV-negative patients, with sensitivity of 87.5% (95% CI = 73.2–95.8%), specificity of 50.3% (95% CI = 42.3–58.3%), PPV 30.7% (95% CI = 22.4–40.0%) and NPV 94.1% (95% CI = 86.8–98.1%)

Nevertheless, a higher cut-off of IP-10 > 150 pg/mL had the highest performance among HIV-positive patients. The cut-off had sensitivity of 79.4% (95% CI = 62.1–91.3%), specificity 47.2% (95% CI = 38.8–55.7%), PPV of 26.5% (95% CI = 18.2–36.1%) and NPV of 90.5% (95% CI = 81.5–96.1%). As shown in Table 2, increasing the cut-off point did not improve the performance of the biomarker.

### 3.3. Performance of CRP and IP-10 Combined

Table 3 shows the performance of the best combinations of CRP and IP-10. At cut-off points of CRP ≥ 10 mg/L and IP-10 > 500 pg/mL the sensitivity and specificity were 92.0% (95% CI = 83.4–97.0%) and 32.9% (95% CI = 27.5–38.6%) and 97.5% (95% CI = 86.8–99.9%).

Among HIV-negative participants, these combinations had a sensitivity of 97.5% (95% CI = 86.8–99.9%) and specificity of 32.9 (95% CI = 27.5–38.6%), while among HIV-positive participants sensitivity and specificity were 83.9 (95% CI = 66.3–94.5%) and 21.1 (95% CI = 14.5–29.0%), respectively.

A CRP cut-off point ≥40 mg/L combined with various IP-10 cut-off points is also shown in Table 3. None of these combinations resulted in a better accuracy of the tests.

Table 4 is a hypothetical scenario of the number of cases that would be correctly diagnosed/excluded by the biomarkers at CRP and IP-10 cut offs when applied to a population of 1000 participants and with a TB prevalence of 21%.

Among HIV-negative patients, a CRP ≥ 10 mg/L (sensitivity 95.3%, specificity 42.6%) would detect 200 out of 210 TB cases. In addition, 453 patients without TB would be CRP-positive and would undergo further TB diagnostic tests. Furthermore, CRP ≥ 10 mg/L and IP-10 > 500 pg/mL would improve the performance, increasing sensitivity from 95.3% to 97.5%, with a stable specificity (42.6% and 43.0%), thereby reducing the number of TB cases missed (from 10 to 5).

Other cut-off points of the biomarkers (either individually or in combination) would result in higher number of TB patients being missed. Also, the table shows that the biomarkers performed fairly well among HIV-negative patients but poorly among HIV-positive patients.

## 4. Discussion

### 4.1. Summary of the Study and Its Findings

In this study, 408 patients with cough of ≥2 weeks were recruited. Of these, 86 (21%) had TB and 46.9% were HIV-positive. CRP at a cut-off point of ≥10 mg/L had the best performance and sensitivity of 91.4% but specificity of 33.2%. The performance was better among HIV-negative patients (sensitivity 95.3%, specificity 42.6%) and worse among HIV-positive patients (sensitivity 84.8%, specificity 22.1%). When CRP cut-offs were increased the sensitivity decreased and specificity increased slightly, as expected.

IP-10 at a cut-off > 100 pg/mL had the best performance. In the whole group the sensitivity was 87.3% and specificity 40.9%. Again, performance was better among HIV-negative patients (sensitivity and specificity 87.5% and 50.3%) than among HIV-positive participants (sensitivity 85.3%, specificity 32.4%). Increasing the cut-off point to IP-10 decreased the sensitivity but increased specificity.

The combination of CRP and IP-10 in the whole study population with CRP ≥ 40 mg/L and IP-10 > 150 pg/mL had sensitivity of 88% and specificity of 46.9%. However, among HIV-negative patients, the optimal cut-off point was CRP ≥ 10 mg/L and IP-10 > 500 pg/mL, which resulted in sensitivity and specificity of 97.5% and 43.0%, respectively. For HIV-positive patients, the optimal cut-off point was CRP > 80 mg/L and IP-10 > 150 pg/mL, with sensitivity of 80.6% and specificity of 42.1%.

### 4.2. Performance of CRP Among Patients with TB Symptoms

We did not find any study that evaluated performance of rapid CRP on a study population comprising HIV-positive and HIV-negative patients. However, a study has shown that the performance of rapid CRP is comparable to that of ELISA (or other lab-based CRP) when used in parallel with each other, with sensitivity of 95% versus 97% and specificity of 51% versus 54% [10]. Hence, we may compare this finding with findings from previous studies that assessed performance of lab-based CRP among populations comprising HIV-positive and HIV-negative patients. The performance of rapid CRP among this study’s population is worse than findings from recent systematic reviews and meta-analyses performed on studies that evaluated performance of the biomarker [11,12,13]. One of the meta-analyses included 17 studies (comprising 5109 participants) and found, at a CRP threshold of 10 mg/L, pooled sensitivity and specificity of 84% (95% CI = 72–91%) and 67% (95% CI = 52–79%), respectively [12]. In another two meta-analyses (each pooled data from ten studies), one found sensitivity of 74% (95% CI = 58–85%) and specificity of 68% (95% CI = 52–80%) [11], and the second study found the sensitivity and specificity to be 86% (95% CI = 80–95%) and 67% (95% CI = 54–79%), respectively [13]. In comparison, this study showed a sensitivity and specificity of 91% (95% CI = 83–97%) and 33% (95% CI = 28–39%), respectively. The difference could be due to different participants’ recruitment criteria/strategies. For instance, we recruited patients with cough of >2 weeks, while some of the studies included in the meta-analyses might have recruited patients with cough of any duration. Also, the studies pooled in the meta-analyses might have used study designs or CRP assay methods different from what we used in our study. These differences could also be due to the sample sizes, methods of TB confirmation and the studies combined in the meta-analyses.

Among HIV-positive patients, CRP performed poorly, which is contrary to findings from three systematic reviews and meta-analyses of studies among HIV-positive patients with TB symptoms. One of the meta-analyses pooled data from studies and reported that a cut-off ≥ 10 mg/L had sensitivity of 93% (95% CI = 90–95%) and specificity of 59% (95% CI = 40–78%) [13]. The second meta-analysis showed that at the cut-off point, the sensitivity and specificity were 87% (95% CI = 76–93%) and 67% (95% CI = 49–81%), respectively [14]. Findings from the third meta-analysis showed a sensitivity of 77% (95% CI = 61–88%) and specificity of 74% (95% CI = 61–83%) [15].

Possible reasons for the difference between our study’s finding and findings from the previous studies may be differences in the studies’ participants’ characteristics, methods of TB confirmation and sample size. Firstly, for instance, some studies might recruit HIV patients irrespective of TB symptoms, but we recruited patients with cough of ≥2 weeks. Hence, it could be that patients in our study had longer symptom duration and, possibly, other opportunistic infections that could have affected the performance of the diagnostics. A meta-analysis [16] and some studies have shown that CRP had poor specificity among patients with TB symptoms who needed admission or self-reported the symptoms (due to high prevalence of pyogenic infections) [17].

However, our study findings are close to findings from a systematic review and meta-analysis which pooled data from twenty-six studies conducted on adult patients in resource-limited settings and showed that CRP at a cut-off point of ≥10 mg/L has sensitivity and specificity of 93% and 40%, respectively [18].

For HIV-negative participants, CRP had better performance. Its sensitivity was good (95.3%), but its specificity was still lower (42.6%) than the WHO-recommended TB screening tool [19]. This finding is corroborated by finding from a prospective study from South Korea, which evaluated the performance of CRP among HIV-negative patients (sensitivity 97% and specificity 41%) [20]. However, findings from another prospective study of HIV-negative patients in China showed lower sensitivity (82%) but higher specificity (60%) [21].

### 4.3. Performance of IP-10 as a Screening Tool

The performance of unstimulated IP-10 was sub-optimal among all categories of patients. Overall, IP-10 had sensitivity of 87.3% and specificity of 40.9%. This is consistent with findings from previous studies. For example, in Uganda, a study of 111 children (comprising 80 HIV-negative and 31 HIV-positive) with presumptive TB and 33 healthy adults (as controls) found that IP-10 performed poorly in differentiating patients with TB and other respiratory diseases (sensitivity 79%, specificity 94%) [22]. In addition, a systematic review and meta-analysis of 14 studies with 2075 participants assessed the diagnostic sensitivity of IP-10, which was 72%, and specificity was 82% [23]. In like manner, a recent systematic review and meta-analysis of six studies showed that IP-10 had sensitivity of 77% (95% CI = 71–83%) and specificity of 84% (95% CI = 80–88%) [8]. However, a prospective study in Norway, of 164 subjects, found that unstimulated IP-10 differentiates active and LTBI irrespective of HIV status [24].

Like CRP, IP-10 performance was better in HIV-negative than HIV-positive patients. The poor performance of IP-10 among HIV-positive patients is supported by a prospective study from Uganda which showed that IP-10 performance was poor in differentiating TB and not TB in HIV-negative and HIV-positive groups [22]. On the other hand, two prospective studies conducted among HIV-negative patients, one from South Korea [25] and the other from the Philippines [26], found that unstimulated IP-10 performed well in differentiating between patients with active TB and LTBI, with sensitivity and specificity of 88% and 91% [25] and 95% and 93% [26], respectively.

To see if using both CRP and IP-10 (one after the other) for TB screening will lead to better performance of the biomarkers as a screening tool for active TB, we assessed the performance of the combination of each cut-off point of CRP with each of the cut-off points of IP-10. Our finding showed that the combination only led to slight improvement in the performance of the biomarkers among HIV-negative patients. It increased the sensitivity by 2.2% (from 95.3% to 97.5%) and the specificity by 0.4% (from 42.6% to 43%). We did not find any study that assessed the performance of combination of the two biomarkers as a screening tool for active TB.

### 4.4. Limitations and Strength of the Study

This study has some limitations. Firstly, the use of solid culture as the reference standard might have missed some TB cases that could be detected by liquid culture, and this might have affected this study’s findings. However, contamination rate could be higher in liquid culture compared to solid culture [27,28,29]. Secondly, we could not compare the performance of the biomarkers to that of the WHO TB symptoms screening because we did not recruit patients irrespective of TB symptoms (we recruited only patients with cough of ≥2 weeks). Lastly, because we diagnosed only pulmonary tuberculosis (PTB), some extra-pulmonary tuberculosis (EPTB) cases might have been missed and this could have caused us to underrate the burden of TB in our study population.

However, this study has some strengths. This study might be the first study from Nigeria that has assessed the performance of CRP and IP-10 as screening tools for active TB. Also, this could be one of the few studies that has compared the performance of the biomarkers between HIV-positive and HIV-negative patients. We also did not find any published previous study that assessed the performance of the combination of CRP and IP-10 as a screening tool for active TB. Hence, this might be the first study on the topic.

## 5. Conclusions

This study showed that close to half of the study population are HIV-infected and 21% of them had TB. In addition, the performance of rapid CRP was poor among HIV-positive patients, fair in the whole study’s population and better among HIV-negative patients. IP-10’s performance was poor among all categories of patients. Diagnostic algorithms that consider testing for CRP as a triage step for TB diagnosis need to include screening for HIV. It is likely that HIV-infected individuals will need to be tested directly by a molecular test such as Xpert, while HIV-negative patients could undergo CRP screening and patients with CRP > 10 mg/L could undergo further tests.

## Figures and Tables

**Figure 1 tropicalmed-10-00306-f001:**
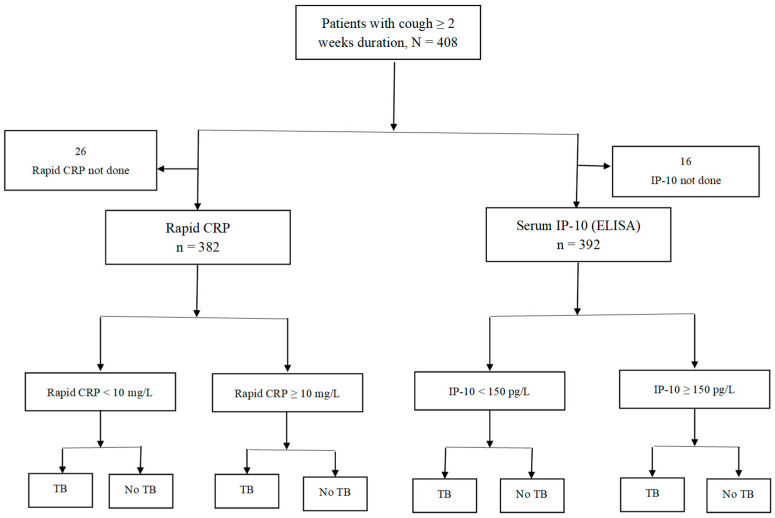
Flow chart.

**Table 1 tropicalmed-10-00306-t001:** Characteristics of participants.

Characteristics	Category	N = 408 n (%)	Missing
Age (Years)	* Median (IQR)	36.0 (28.5–43.0)	
Gender	Male	194 (47.5)	
	Female	214 (52.5)	
Cough	Yes	408 (100)	
Cough duration (weeks)	* Median (IQR)	4.0 (2.0–6.5)	
Nose bleeding	Yes	56 (13.7)	
Fever	Yes	271 (66.4)	
Weight loss	Yes	280 (68.6)	
Night sweats	Yes	210 (51.5)	
Chest pain	Yes	256 (62.7)	
Body weakness	Yes	233 (57.1)	
Loss of appetite	Yes	140 (34.3)	
Other illness	Yes	55 (13.5)	
HIV status	Negative	206 (53.1)	20
	Positive	182 (46.9)	
Previous TB treatment	None	348 (85.3)	
	Previous	58 (14.2)	
	Current	2 (0.5)	
If previously treated, treatment class	Relapse	32 (53.3)	
	Retreatment	2 (3.3)	
	Not known	26 (43.3)	
Months since last treatment	* Median (IQR)	24.0 (7.0–58.0)	
Culture performed	Yes	396 (97)	
Culture results	Negative	326 (82.3)	
	Positive	62 (15.7)	
	Contaminated	8 (2.0)	
Xpert	Xpert MTB/RIF	400 (98.0)	
	Ultra Xpert	7 (1.7)	
	Not performed	1 (0.2)	
Xpert result	MTB not detected	329 (80.8)	
	MTB detected	77 (19.0)	
	Invalid	1 (0.2)	
Xpert grading	Very low	10 (13.0)	
	Low	16 (20.8)	
	Medium	23 (29.9)	
	High	28 (36.3)	
Xpert RIF resistance	Not detected	70 (90.9)	
	Detected	7 (9.1)	
Confirmed TB	No	322 (78.9)	
	Yes	86 (21.1)	
CRP rapid test performed		382 (93.6)	26
CRP rapid test result	<10,000 µ g/L	107 (28.0)	
	10,000–40,000 µ g/L	126 (33.0)	
	40,000–80,000 µ g/L	38 (9.9)	
	>80,000 µ g/L	111 (29.1)	
IP-10 result (pg/mL)	* Median (IQR)	148 (73–449)	16

***** Number and percentage unless specified, IQR = Interquartile Range.

**Table 2 tropicalmed-10-00306-t002:** Performance of CRP and IP-10 as a screening tool for active TB.

Population	Test	Cut-Off	Sensitivity % (95% CI)	Specificity % (95% CI)	PPV % (95% CI)	NPV % (95% CI)
All	CRP rapid	>10 mg/L	91.4 (83.0–96.5)	33.2 (27.9–38.9)	26.9 (21.8–32.6)	93.5 (87.0–97.3)
		>40 mg/L	72.8 (61.8–82.1)	70.1 (64.6–75.2)	39.6 (31.7–47.9)	90.6 (86.1–94.0)
		>80 mg/L	61.7 (50.3–72.3)	79.7 (74.7–84.1)	45.0 (35.6–54.8)	88.6 (84.2–92.1)
	IP-10	>50 pg/ml	94.9 (87.5–98.6)	17.6 (13.5–22.2)	22.5 (18.1–27.4)	93.2 (83.5–98.1)
		>100 pg/mL	87.3 (78.0–93.8)	40.9 (35.4–46.6)	27.2 (21.8–33.1)	92.8 (87.1–96.5)
		>150 pg/mL	79.7 (69.2–88.0)	58.1 (52.5–63.7)	32.5 (25.9–39.6)	91.9 (87.2–95.3)
		>200 pg/mL	75.9 (65.0–84.9)	65.5 (59.9–70.8)	35.7 (28.5–43.5)	91.5 (87.1–94.8)
		>250 pg/mL	69.6 (58.2–79.5)	70.6 (65.2–75.6)	37.4 (29.6–45.8)	90.2 (85.8–93.6)
		>300 pg/mL	62.0 (50.4–72.7)	75.7 (70.6–80.4)	39.2 (30.6–48.3)	88.8 (84.3–92.3)
HIV −ve	CRP rapid	>10 mg/L	95.3 (84.2–99.4)	42.6 (34.7–50.8)	31.5 (23.7–40.3)	97.1 (89.8–99.6)
		>40 mg/L	76.7 (61.4–88.2)	75.5 (67.9–82.0)	46.5 (34.5–58.7)	92.1 (86.0–96.2)
		>80 mg/L	62.8 (46.7–77.0)	81.9 (75.0–87.6)	49.1 (35.4–62.9)	88.8 (82.5–93.5)
	IP-10	>50 pg/mL	95.0 (83.1–99.4)	23.9 (17.5–31.3)	23.9 (17.5–31.3)	95.0 (83.1–99.4)
		>100 pg/mL	87.5 (73.2–95.8)	50.3 (42.3–58.3)	30.7 (22.4–40.0)	94.1 (86.8–98.1)
		>150 pg/mL	80.0 (64.4–90.9)	69.2 (61.4–76.3)	39.5 (28.8–51.0)	93.2 (87.1–97.0)
		>200 pg/mL	75.0 (58.8–87.3)	77.4 (70.1–83.6)	45.5 (33.1–58.2)	92.5 (86.6–96.3)
		>250 pg/mL	67.5 (50.9–81.4)	81.1 (74.2–86.9)	47.4 (34.0–61.0)	90.8 (84.9–95.0)
		>300 pg/mL	57.5 (40.9–73.0)	85.5 (79.1–90.6)	50.0 (34.9–65.1)	88.9 (82.8–93.4)
HIV +ve	CRP rapid	>10 mg/L	84.8 (68.1–94.9)	22.1 (15.4–30.0)	20.9 (14.4–28.8)	85.7 (69.7–95.2)
		>40 mg/L	66.7 (48.2–82.0)	64.7 (56.1–72.7)	31.4 (20.9–43.6)	88.9 (81.0–94.3)
		>80 mg/L	57.6 (39.2–74.5)	78.7 (70.8–85.2)	39.6 (25.8–54.7)	88.4 (81.3–93.5)
	IP-10	>50 pg/mL	94.1 (80.3–99.3)	11.3 (6.6–17.7)	20.3 (14.3–27.4)	88.9 (65.3–98.6)
		>100 pg/mL	85.3 (68.9–95.0)	32.4 (24.8–40.8)	23.2 (16.1–31.6)	90.2 (78.6–96.7)
		>150 pg/mL	79.4 (62.1–91.3)	47.2 (38.8–55.7)	26.5 (18.2–36.1)	90.5 (81.5–96.1)
		>200 pg/mL	76.5 (58.8–89.3)	51.4 (42.9–59.9)	27.4 (18.7–37.5)	90.1 (81.5–95.6)
		>250 pg/mL	70.6 (52.5–84.9)	58.5 (49.9–66.7)	28.9 (19.5–39.9)	89.2 (81.1–94.7)
		>300 pg/mL	64.7 (46.5–80.3)	64.1 (55.6–72.0)	30.1 (19.9–42.0)	88.3 (80.5–93.8)

CRP = C-reactive protein, IP-10 = Interferon-Gamma-inducible protein 10, PPV = Positive predictive value, NPV = Negative predictive value, 95% CI = 95% Confidence Interval, HIV −ve = HIV-negative, HIV +ve = HIV-positive.

**Table 3 tropicalmed-10-00306-t003:** Performance of CRP and IP-10 as a combined screening tool for active TB.

Rapid CRP Cut-Off	IP-10 Cut-Off	Sensitivity (95% CI)			Specificity (95% CI)		
		All	HIV −ve	HIV +ve	All	HIV −ve	HIV +ve
>10 mg/L	>50 pg/mL	98.7 (92.8–100)	100 (91.2–100)	96.8 (83.3–99.9)	10.6 (7.3–14.7)	15.2 (9.9–22.0)	5.3 (2.1–10.5)
	>100 pg/mL	97.3 (90.7–99.7)	100 (91.2–100)	93.5 (786–99.2)	20.2 (15.8–25.3)	26.5 (19.6–34.3)	13.5 (8.2–20.5)
	>150 pg/mL	96.0 (88.8–99.2)	100 (91.2–100)	90.3 (74.2–98.0)	27.4 (22.4–32.9)	37.1 (29.4–45.3)	16.5 (10.7–24.0)
	>200 pg/mL	93.3 (85.1–97.8)	97.5 (86.8–99.9)	87.1 (70.2–96.4)	27.7 (22.7–33.3)	37.7 (30.0–46.0)	16.5 (10.7–24.0)
	>250 pg/mL	92.0 (83.4–97.0)	97.5 (86.8–99.9)	83.9 (66.3–94.5)	28.8 (23.6–34.3)	39.1 (31.2–47.3)	17.3 (11.3–24.8)
	>300 pg/mL	92.0 (83.4–97.0)	97.5 (86.8–99.9)	83.9 (66.3–94.5)	29.8 (24.6–35.4)	40.4 (32.5–48.7)	18.0 (11.9–25.6)
	>350 pg/mL	92.0 (83.4–97.0)	97.5 (86.8–99.9)	83.9 (66.3–94.5)	30.8 (25.6–36.5)	42.4 (34.4–50.7)	18.0 (11.9–25.6)
	>400 pg/mL	92.0 (83.4–97.0)	97.5 (86.8–99.9)	83.9 (66.3–94.5)	31.2 (25.9–36.8)	42.4 (34.4–50.7)	18.0 (11.9–25.6)
	>450 pg/mL	92.0 (83.4–97.0)	97.5 (86.8–99.9)	83.9 (66.3–94.5)	31.2 (25.9–36.8)	42.4 (34.4–50.7)	18.0 (11.9–25.6)
	>500 pg/mL	92.0 (83.4–97.0)	97.5 (86.8–99.9)	83.9 (66.3–94.5)	32.9 (27.5–38.6)	43.0 (35.0–51.3)	21.1 (14.5–29.0)
>40 mg/L	>50 pg/mL	94.7 (86.9–98.5)	95.0 (83.1–99.4)	93.5 (78.6–99.2)	14.7 (10.9–19.3)	20.5 (14.4–99.4)	8.3 (4.2–99.2)
	>100 pg/mL	90.7 (81.7–96.2)	92.5 (79.6–98.4)	87.1 (70.2–96.4)	32.2 (26.9–37.9)	40.4 (32.5–48.7)	24.1 (17.1–32.2)
	>150 pg/mL	88.0 (78.4–94.4)	92.5 (79.6–98.4)	80.6 (62.5–92.5)	46.9 (41.1–52.8)	56.3 (48.0–64.3)	37.6 (29.3–46.4)
	>200 pg/mL	85.3 (75.3–92.4)	90.0 (76.3–97.2)	77.4 (58.9–90.4)	51.0 (45.1–56.9)	62.3 (54.0–70.0)	39.1 (30.8–47.9)
	>250 pg/mL	82.7 (72.2–90.4)	87.5 (73.2–95.8)	74.2 (55.4–88.1)	53.8 (47.9–59.6)	64.9 (56.7–72.5)	42.1 (33.6–51.0)
	>300 pg/mL	81.3 (70.7–89.4)	85.0 (70.2–94.3)	74.2 (55.4–88.1)	56.2 (50.3–61.9)	66.9 (58.8–74.3)	44.4 (35.8–53.2)
	>350 pg/mL	81.3 (70.7–89.4)	85.0 (70.2–94.3)	74.2 (55.4–88.1)	58.6 (52.7–64.3)	70.2 (62.2–77.4)	45.9 (37.2–54.7)
	>400 pg/mL	81.3 (70.7–89.4)	85.0 (70.2–94.3)	74.2 (55.4–88.1)	59.6 (53.7–65.3)	71.5 (63.6–78.6)	45.9 (37.2–54.7)
	>450 pg/mL	81.3 (70.7–89.4)	85.0 (70.2–94.3)	74.2 (55.4–88.1)	59.6 (53.7–65.3)	71.5 (63.6–78.6)	45.9 (37.2–54.7)
	>500 pg/mL	76.0 (64.7- 85.1)	80.0 (64.4–90.9)	67.7 (48.6–83.3)	68.8 (63.2–74.1)	74.2 (66.4–80.9)	63.2 (54.4–71.4)
>80 mg/L	>50 pg/mL	94.7 (86.9–98.5)	95.0 (83.1–99.4)	93.5 (78.6–99.2)	16.4 (12.4–98.5)	21.9 (15.5–29.3)	10.5 (5.9–17.0)
	>100 pg/mL	89.3 (80.1–95.3)	90.0 (76.3–97.2)	87.1 (70.2–96.4)	36.0 (30.5–41.8)	45.0 (36.9–53.3)	27.1 (19.7–35.5)
	>150 pg/mL	85.3 (75.3–92.4)	87.5 (73.2–95.8)	80.6 (62.5–92.5)	51.4 (45.5–57.2)	60.9 (52.7–68.8)	42.1 (33.6–51.0)
	>200 pg/mL	81.3 (70.7–89.4)	82.5 (67.2–92.7)	77.4 (58.9–90.4)	56.5 (50.6–62.3)	67.5 (59.5–74.9)	45.1 (36.5–54.0)
	>250 pg/mL	77.3 (66.2–86.2)	77.5 (61.5–89.2)	74.2 (55.4–88.1)	60.6 (54.8–66.3)	70.9 (62.9–78.0)	50.4 (41.6–59.2)
	>300 pg/mL	76.0 (64.7–85.1)	75.0 (58.8–87.3)	74.2 (55.4–88.1)	63.7 (57.9–69.2)	72.8 (65.0–79.8)	54.1 (45.3–62.8)
	>350 pg/mL	76.0 (64.7–85.1)	75.0 (58.8–87.3)	74.2 (55.4–88.1)	66.1 (60.4–71.5)	76.2 (68.6–82.7)	55.6 (46.8–64.2)
	>400 pg/mL	76.0 (64.7–85.1)	75.0 (58.8–87.3)	74.2 (55.4–88.1)	67.5 (61.8–72.8)	78.1 (70.7–84.5)	55.6 (46.8–64.2)
	>450 pg/mL	76.0 (64.7–85.1)	75.0 (58.8–87.3)	74.2 (55.4–88.1)	67.5 (61.8–72.8)	78.1 (70.7–84.5)	55.6 (46.8–64.2)
	>500 pg/mL	65.3 (53.5–76.0)	67.5 (50.9–81.4)	58.1 (39.1–75.5)	78.8 (73.6–83.3)	80.8 (73.6–86.7)	77.4 (69.4–84.2)

CRP = C-reactive protein, IP-10 = Interferon-Gamma-inducible protein 10 95% CI = 95% Confidence Interval, HIV −ve = HIV-negative, HIV +ve = HIV-positive.

**Table 4 tropicalmed-10-00306-t004:** Hypothetical number of patients that would be selected for further TB tests using CRP and IP-10 as TB screening tools with a TB prevalence of 21% among 1000 presumptive TB cases.

				Bact. Positive, n = 210		Bact. Negative, n = 790	
		Sensitivity	Specificity	Correctly Selected	Missed	Correctly Excluded	Included for Further Tests
All	CRP > 10 mg/L	91.4%	33.2%	192	18	262	528
	>40 mg/L	72.8%	70.1%	153	57	554	336
	>80 mg/L	61.7%	79.7%	130	80	630	160
	IP-10 > 100 pg/mL	87.3%	40.9%	183	27	323	467
	>150 pg/L	79.7%	58.1%	167	143	459	331
	CRP > 10 mg/L and IP-10 > 500 pg/mL	92.0%	32.9%	193	17	260	530
	CRP > 40 mg/l and IP-10 > 150 pg/mL	88.0%	46.9%	185	25	371	419
	CRP > 40 mg/L and IP-10 > 450 pg/mL	81.3%	59.6%	171	39	471	319
	CRP > 80 mg/L and IP-10 > 100 pg/mL	89.3%	36.0%	188	22	284	506
	CRP > 80 mg/L and IP-10 > 150 pg/mL	85.3%	51.4%	179	31	406	384
HIV −ve	CRP > 10 mg/L	95.3%	42.6%	200	10	337	453
	>40 mg/L	76.7%	75.5%	161	49	596	194
	>80 mg/L	62.8%	81.9%	132	78	647	143
	IP-10 > 100 pg/mL	87.5%	50.3%	184	26	397	393
	>150 pg/mL	80.0%	69.2%	168	42	547	243
	CRP > 10 mg/L and IP-10 > 500 pg/mL	97.5%	43.0%	205	5	340	450
	CRP > 40 mg/L and IP-10 > 150 pg/mL	92.5%	56.3%	194	16	445	345
	CRP > 40 mg/L and IP-10 > 200 pg/mL	90.0%	62.3%	189	21	492	298
	CRP > 40 mg/l and IP-10 > 450 pg/mL	85.0%	71.5%%	179	31	565	225
	CRP > 40 mg/L and IP-10 > 500 pg/mL	80.0%	74.2%	168	42	586	204
	CRP > 80 mg/L and IP-10 > 100 pg/mL	90.0%	45.0%	189	21	356	434
	CRP > 80 mg/L and IP-10 > 150 pg/mL	87.5%	60.9%	184	26	481	309
HIV +ve	CRP > 10 mg/L	84.8%	22.1%	178	32	175	615
	>40 mg/L	66.7%	64.7%	140	70	511	279
	>80 mg/L	57.6%	78.7%	121	89	622	168
	IP-10 > 100 pg/mL	85.3%	32.4%	179	31	256	534
	>150 pg/mL	79.4%	47.2%	167	43	373	417
	CRP > 10 mg/L and IP-10 > 500 pg/mL	83.9%	21.1%	176	34	167	623
	CRP > 40 mg/L and IP-10 > 150 pg/mL	80.6%	37.6%	169	41	297	493
	CRP > 80 mg/L and IP-10 > 150 pg/mL	80.6%	42.1%	169	41	333	457

## Data Availability

The raw data supporting the conclusions of this article will be made available by the authors on request.

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
