# Peer review of "Evaluation of Performance of C-Reactive Protein (CRP) and Interferon-Gamma-Inducible Protein 10 (IP-10) as Screening for Active Tuberculosis"

_tropicalmed, 2025, doi:10.3390/tropicalmed10110306_

Round 1
Reviewer 1 Report
Comments and Suggestions for Authors
- Table 1 is too crowded, just write for the 'Yes'
- Narration for the table redundant with the table itself, please make a comment for the important findings.
- Where The number of bacteria on the top of table 4 come from? Please explain
- Did the author analyse all TB cases, Rifampicin sensitive and resistant? Wil it make difference in the results?

Author Response
Reviewer 1
Comments and Suggestions for Authors
Comment 1: Table 1 is too crowded, just write for the 'Yes'
Response 1: Thanks for pointing this out. We agree with this comment. Therefore, we have made the suggested correction on the table.
Comment 2: Narration for the table redundant with the table itself, please make a comment for the important findings.
Response 2: Thank you once again, for pointing this out. We agree with this comment. We have now reduced the narration of the table, to highlight only important findings. The changes made can be find below table 1, from line 180 - 190
Comment 3: Where did the number of bacteria on the top of table 4 come from? Please explain
Response 3: Thanks for pointing this out. We agree with this comment. Please, be informed that the numbers of bacteria-positive and bacteria-negative cases displayed on table were obtained based on a prevalence of 21% (which is the patients who had active TB in this study) from a hypothetical population of 1000. We then used this to show the numbers of TB cases that will be correctly diagnosed or missed at various sensitivity and specificity.
Comment 4: Did the author analyze all TB cases, Rifampicin sensitive and resistant? Wil it make difference in the results?
Response 4: Thanks for pointing this out. We agree with this comment. In this study, we analyzed all the TB cases. Regarding the second question, though, it is said that the serum level of C-reactive protein increases in Rifampicin resistance, I don’t think disaggregating the data into Rifampicin-resistant and Rifampicin-sensitive will significantly change the findings from this study, because the proportion of patients/participants with Rifampicin resistance in this study was less than 10%.
Reviewer 2 Report
Comments and Suggestions for Authors
The work is methodologically good, but there are small revisions to be made in the writing of the article:
PLHIV - please add the full significance firstly and in the abstract.
line 82 - " we evaluated.." , please revise all sentences written in the first person. In scientific writing, sentences should not be written this way. Furthermore, the last paragraph of the introduction should directly explain the objectives of this work.
DOTS clinics ? Please revise carefully all the abbreviations
I think some of the information in the results is confusing and leaves the authors lost. All statistical data should be presented in tables for ease of reading. There is no need to repeat the information in the table and text.
Future? It was important for the authors to dedicate a paragraph to the steps and tests to be performed and studied in the future.
REFERENCES: All references are prior to 2020, that is, all are more than 5 years old. Please reflect and incorporate updated references into your work.
Author Response
Reviewer 2
Comments and Suggestions for Authors
The work is methodologically good, but there are small revisions to be made in the writing of the article:
Comment 1: PLHIV - please add the full significance firstly and in the abstract.
Response 1: Thanks for pointing this out. It is not clear if the significance mentioned above implies abbreviating PLHIV, without first writing it in full. However, we have fully written PLHIV with the abbreviation in a bracket, at its first appearance in this manuscript. Find this correction on page 1. line 13.
Comment 2: line 82 - " we evaluated.." , please revise all sentences written in the first person. In scientific writing, sentences should not be written this way. Furthermore, the last paragraph of the introduction should directly explain the objectives of this work.
Response 2: Thanks for pointing this out. We agree with the first aspect of the comment, but its second aspect had already been addressed in the manuscript. Hence, we have made the correction suggested on line 82. However, I think the last paragraph of the introduction already has the objective of the study. This can be found on page 2, lines 84 – 86.
Comment 3: DOTS clinics? Please revise carefully all the abbreviations
Response 3: Thanks for this observation. We agree with this comment. We have written in full, the abbreviation “DOTs”, with the abbreviation in a bracket, at its first appearance in this manuscript. In addition, we have properly written the abbreviation “DOTs” instead of “DOTS” that we initially wrote everywhere “DOTs” appears in the manuscript.
Comment 4; I think some of the information in the results is confusing and leaves the authors lost. All statistical data should be presented in tables for ease of reading. There is no need to repeat the information in the table and text.
Response 4: Thanks for pointing this out. We agree with this comment. Consequently, the narration of the tables has now been reduced to highlight only important findings. To confirm this, please, see page 6, lines 182 – 190, and pages 7 and 8, lines 199 – 253.
Comment 5: Future? It was important for the authors to dedicate a paragraph to the steps and tests to be performed and studied in the future.
Response 5: This comment is noted
Comment 6: REFERENCES: All references are prior to 2020, that is, all are more than 5 years old. Please reflect and incorporate updated references into your work.
Response 6: Thanks for pointing this out. We strongly agree with this comment. Therefore, references have been updated with recent relevant studies. These changes or corrections can be seen across the manuscript’s texts and in the bibliography section of the manuscript.
Reviewer 3 Report
Comments and Suggestions for Authors
Major concern
Similar literature is already available
[https://pubmed.ncbi.nlm.nih.gov/25719208/]
[https://www.frontiersin.org/journals/tuberculosis/articles/10.3389/ftubr.2024.1377540/full]
[Acute phase proteins and IP-10 as triage tests for the diagnosis of tuberculosis: systematic review and meta-analysis - PubMed]
then what is the purpose of this investigation.
Did the authors find any significant changes in HIV patients.
Abstract
There should be novelty in the abstract.
Introduction
Its very general. This section should focus on the problem statement.
There are many abbreviations which have not been defined.
Methodology
Section 2.2, the critical concentration of drug should be provided along with the duration of treatment.
The inclusion criteria are not clear
The exclusion criteria is also not clear.
Results
Table 1, is the sign/symptoms are necessary in this table. They are generally not required. Did the authors find a difference in sign/symptoms?
What is the CRP and IFN normal level?
Table 2 may be supplemented with some kind of association.
Lines 192-232 should be revised and only major values should be provided. The results not explained enough in this study.
Table 4 caption should be revised.
Discussion section should be discussed the current study finding and the major difference and performance with previous ones.
Conclusion should also be revised as per the current study findings
Author Response
Reviewer 3
Comments and Suggestions for Authors
Major concern
Comment 1: Similar literature is already available
[https://pubmed.ncbi.nlm.nih.gov/25719208/]
[https://www.frontiersin.org/journals/tuberculosis/articles/10.3389/ftubr.2024.1377540/full]
[Acute phase proteins and IP-10 as triage tests for the diagnosis of tuberculosis: systematic review and meta-analysis - PubMed]
then, what is the purpose of this investigation?
Response 1: Thanks for pointing this out, but we partially agree with this comment Because, though, there were similar studies published, none of the studies mentioned by the reviewer has data from Nigeria so we feel this study is useful by exploring these questions in a different context.
Comment 2: Did the authors find any significant changes in HIV patients?
Response 2: Thanks for your question. We found out that the performance of the biomarkers were generally poor, but it was better among HIV-negative compared to HIV-positive patients
Abstract
Comment 3: There should be novelty in the abstract.
Response 3: Thanks for the comment made above, but we strongly disagree with the comment. We believe that there is novelty in the abstract, because as stated in the abstract, most of the available TB diagnostics are sputum-based and these are not performing well among HIV patients. Therefore, this study evaluated two non-sputum-based diagnostics with a view to seeing if they can be useful for TB diagnosis among the patients.
Introduction
Comment 4: It’s very general. This section should focus on the problem statement.
Response 4: Thanks for the above comment out. However, we also strongly disagree with this comment The introduction section of this manuscript specifically addresses the research problem.
Comment 5: There are many abbreviations which have not been defined.
Response 5: Thanks for pointing this out. We agree with this comment. In response to the comment, we have written every abbreviation in full at its first appearance in this manuscript.
Methodology
Comment 6: Section 2.2, the critical concentration of drug should be provided along with the duration of treatment.
Response 6: Thanks for the above comment. However, we strongly disagree with the comment. Because, the reviewer’s comment on section 2.2 under methodology, is not done in Nigeria or applicable in the country. Please, note that in this section, we described what is done in Nigeria.
Comment 7: The inclusion criteria are not clear
Comment 8: The exclusion criteria is also not clear.
Response 7 and Response 8: Thanks for the above comments. However, we strongly disagree with the comments. Because, inclusion and exclusion criteria were in the manuscript as stated below;
- Inclusion criteria
- Age ≥ 18 years
- Cough of ≥ 2 weeks
- Consented to participate
- Agreed to provide the needed specimens’ samples (blood and sputum)
- 5. Exclusion criteria
- Currently on TB treatment
- Already diagnosed of TB but has not commenced TB
- Patients with language barriers and no appropriate interpreter
- Patients willing to participate but could not read/write and no legally assigned representative to sign consent form and act as a witness
Please, confirm our claim on pages 3 and 4, lines 131 – 141 in the manuscript.
Results
Comment 9: Table 1, is the sign/symptoms are necessary in this table. They are generally not required. Did the authors find a difference in sign/symptoms?
Response 9: We think the symptoms needed to be presented, because data on more than the four main TB symptoms were collected.
Comment 10: What is the CRP and IFN normal level?
Response 10: Normal level of CRP is < 1.0 mg/dL or 10.0 mg/L, while the normal level of IP-10 = 150 – 200 pg/ml
Comment 11: Table 2 may be supplemented with some kind of association.
Response 11: This suggestion by the reviewer may not be needed, because assessment of any kind of association was not one of the objectives of this study, and we analyzed this study’s data based on the objectives.
Comment 12: Lines 192-232 should be revised and only major values should be provided. The results not explained enough in this study.
Response 12: Thanks for pointing this out. We agree with this comment. Consequently, the reviewer’s suggestion on lines 192-232 has been acted upon, by ensuring that only major and relevant results were reported under or after the tables. Please, see lines 199 – 253, on pages 7 and 8, to confirm our claim.
Comment 13: Table 4 caption should be revised.
Response 13: Thanks for pointing this out. We agree with this comment. Hence, the caption for the table has been revised as suggested. Kindly see lines 276 and 277, on page 10, for confirmation of our claim.
Comment 14: Discussion section should be discussed the current study finding and the major difference and performance with previous ones.
Response 14: Thanks for pointing this out. We agree with this comment. Therefore, the section has been updated with recent studies’ findings accordingly. Please, lines 317 – 387 on pages 11 and 12, lines 402 – 404 on page 13, and the bibliography section of the manuscript.
Comment 15: Conclusion should also be revised as per the current study findings.
Response 15: We have updated the discussion and conclusion sections with findings from recent relevant studies
Round 2
Reviewer 3 Report
Comments and Suggestions for Authors
The authors adressed all our commments